# D-GAN: Divergent generative adversarial approach for positive-unlabeled learning and counter-examples generation

## Abstract

Positive Unlabeled (PU) learning consists in learning to distinguish samples of our class of interest, the positive class, from the counter-examples, the negative class, by using positive labeled and unlabeled samples during the training. Recent approaches exploit the GANs abilities to address the PU learning problem by generating relevant counter-examples. In this paper, we propose a new GAN-based PU learning approach named Divergent-GAN (D-GAN). The key idea is to incorporate a standard Positive Unlabeled learning risk inside the GAN discriminator loss function. In this way, the discriminator asks the generator to converge towards the unlabeled samples distribution while diverging from the positive samples distribution. This enables the generator convergence towards the unlabeled counter-examples distribution without using prior knowledge, while keeping the standard adversarial GAN architecture. In addition, we discuss normalization techniques in the context of the proposed framework. Experimental results show that the proposed approach overcomes previous GAN-based PU learning methods issues, and it globally outperforms two-stage state of the art PU learning performances in terms of stability and prediction on both simple and complex image datasets.

## 1 Introduction

Nowadays, the number of available labeled datasets dedicated to perception applications has considerably augmented (Russakovsky et al., 2015), (Yu et al., 2015), (Cordts et al., 2016). However, when learning methods trained on these datasets are applied on real data, their performances are likely to deteriorate. Consequently, it is necessary to use a dataset specialized for the given target application. It turns out that it can be easy to get unlabeled data in some applications domains such as autonomous driving. Positive Unlabeled (PU) learning enables to use these unlabeled data such that they are combined with labeled samples of our class of interest: the positive class. The interest is that unlabeled data can contain relevant counter-examples: the negative samples. The difficulty is that unlabeled data can also contain a fraction $\pi_p$ of unlabeled positive samples. (Sansone et al., 2018) enumerates several learning problems which can be addressed in this way like the information retrieval and the One-vs-Rest task.

Several PU learning methods exist and some of them are adapted for image classification. They are generally classified into these two categories: - censoring PU learning, formalized by Elkan & Noto (2008) and recently improved by Northcutt et al. (2017); - case-control PU learning, introduced by Ward et al. (2009), formalized by du Plessis et al. (2014), and then consecutively improved by Du Plessis et al. (2015) and Kiryo et al. (2017). However we focus our attention in this article on the recently proposed GAN-based PU approaches. So we classify PU learning approaches into the two following groups suggested by Kiryo et al. (2017): one-stage and two-stage PU methods. One-stage ones such as the unbiased PU method (uPU) (Du Plessis et al., 2015) and the non-negative PU method (nnPU) (Kiryo et al., 2017) consist in training a classifier directly with the PU dataset. These methods have the advantage to need only one training of the classifier. But they require dataset prior knowledge and consequently uPU and nnPU need to be combined with an approach estimating the prior knowledge (Jain et al., 2016), (Ramaswamy et al., 2016), (Christoffel et al., 2016). On the other hand, two-stage PU methods prepare during the first stage a PN dataset. For example, Rank Pruning method (RP) (Northcutt et al., 2017) realizes several consecutive trainings of a binary

classifier. After each training, only the samples considered as the most confident are kept in order to substitute the unlabeled samples for the next training of the same classifier. RP achieves two-stage state of the art performances without prior knowledge, but with a high training computational cost and is limited in its prediction performances on complex image datasets like CIFAR-10.

Recently, a new subcategory of two-stage PU methods appeared: GAN-based PU methods. They address the PU learning challenge by producing, thanks to an adversarial training (Goodfellow et al., 2014), relevant generated samples from a PU dataset during the first step. Then they are used to train a standard Positive Negative (PN) classifier during the second step. Existing GAN-based PU methods PGAN (Chiaroni et al., 2018) and GenPU (Hou et al., 2018) are presented and their issues are discussed in the next section 2. This article addresses these issues with a new GAN-based PU learning approach: the Divergent-GAN. Our contributions are the following:

- we propose to incorporate a PU learning risk inside the standard GAN (Goodfellow et al., 2014) discriminator loss function. The intuition behind is to have the generative model solving the PU learning problem formulated in the discriminator loss function. In this way the generator learns the distribution of the samples which are both unlabeled and not positive: the negative ones included in the unlabeled dataset;

- (side contribution) we highlight a Batch Normalization (BN) (Ioffe & Szegedy, 2015) critical issue: A learning model which manipulates different minibatches distributions should not use BN. Alternative normalization techniques are discussed (section 3.2) and experimented (section 4.1) in the context of the proposed framework which manages positive, unlabeled and generated minibatches.

Thanks to this new framework, we address the previous GAN-based PU methods main issues: It reduces the overfitting problem of the PGAN method, and it learns the counter-examples distribution included in the unlabeled dataset without using prior knowledge while keeping a standard GAN architecture. Consequently, the D-GAN globally outperforms two-stage state of the art PU learning performances on simple MNIST and complex CIFAR-10 image datasets.

From a practical point of view, several previous PU methods benefits are combined in the proposed framework as listed in Table 1. The D-GAN code is available at [1].

The article is structured as follow. Section 2 presents GAN-based PU learning approaches. Section 3 describes the proposed method. Section 4 presents the corresponding experimental results. Then in section 5, we finish by a conclusion where we also discuss perspectives.

## 2 RELATED WORK

GAN-based PU learning approaches are a recent subcategory of two-stage PU methods, as proposed in PGAN (Chiaroni et al., 2018) and GenPU (Hou et al., 2018). The interest of using GANs is double: GANs enable relevant data augmentation and they use high-level feature metrics to evaluate generated samples quality, thanks to the use of a discriminator. The generated samples replace the unlabeled ones by learning on the latter (Chiaroni et al., 2018), or on both unlabeled and positive labeled ones as (Hou et al., 2018) during the first step. Both methods exploit GANs (Goodfellow et al., 2014) benefits, but they are not suitable under the same datasets conditions and the reasonings are different:

- GenPU method is based on the original GAN convergence (Goodfellow et al., 2014), such that: $\pi_P \cdot p_{G_P} + (1 - \pi_P) \cdot p_{G_N} \to p_U$, with $p_{G_P}$ the distribution of positive samples generated by the generator $G_P$, $p_{G_N}$ the distribution of the negative samples generated by $G_N$, and $p_U$ the distribution of real unlabeled samples. The same idea can be expressed by using one single generator $G_N$ such that we replace the generated positive samples by the positive labeled samples initially present in a PU dataset. In practice, GenPU seems to be an interesting PU method on simple datasets when we own few positive labeled samples, and it generates relevant counter-examples. However, training adversarially five learning models instead of two to address standard PU learning challenge is not necessary and more computational demanding. Moreover, this amplifies the mode collapse issue, and GenPU

---

[1]The code will be publicly available after the anonymous reviewing period.

training optimization functions need three additional hyper-parameters combined with prior knowledge. This is not practical.

- PGAN method is trained to converge towards the unlabeled dataset distribution during the first step. During the second step, it exploits GANs imperfections such that the generated distribution at the adversarial equilibrium is still separable from the unlabeled samples distribution by a classifier. It presents a relatively steadier behaviour and better prediction performances than the two-stage baseline RP method on the complex RGB image dataset CIFAR-10 without prior knowledge. However, it is less suitable for relatively simpler datasets like MNIST. The generated samples are all considered as negative samples by the classifier. But this is possible only if the generated samples distribution converges close enough towards the unlabeled samples distribution, while not matching with it. So if the PGAN first stage works as expected theoretically by Goodfellow et al. (2014), then the PGAN classification second stage falls back into the initial PU learning problem. We highlight this first stage overfitting problem in figure 5(b).

The proposed approach overcomes previous enumerated GAN-based PU methods issues, while merging their respective advandages to address the standard PU learning task. This is achieved with the novel reasoning presented in the next section.

Table 1: Practical D-GAN proposed approach advantages.

| Methods | D-GAN (proposed) | PGAN (Chiaroni et al., 2018) | GenPU (Hou et al., 2018) | RP (Northcutt et al., 2017) | nnPU (Kiryo et al., 2017) | uPU (Du Plessis et al., 2015) |
|---|---|---|---|---|---|---|
| No need of priori knowledge | ✓ | ✓ | × | ✓ | × | × |
| Practical PU framework | ✓ | ✓ | × | ✓ | ✓ | ✓ |
| Simple PU dataset analysis | ✓ | × | ✓ | ✓ | ✓ | ✓ |
| Complex PU dataset analysis | ✓ | ✓ | × | × | | |
| Generation of relevant counter-examples | ✓ | × | ✓ | | | |
| Noisy Labeled learning (section 6.3) | ✓ | × | × | ✓ | × | × |

## 3 PROPOSED METHOD

This section starts by discussing some particularities of a standard PU learning risk (equation 1). Then we propose to incorporate this risk into a generic GAN framework in order to guide the generator convergence towards the distribution of the negative samples included in the unlabeled dataset. In addition, we discuss normalization techniques in the context of this framework which manipulates three different types of minibatches: positive, unlabeled and generated ones.

### 3.1 A DIVERGENT APPROACH FROM THE DISCRIMINATOR POINT OF VIEW

The D-GAN intuition can be expressed as follow. The discriminator $D$ addresses to the generator $G$ the riddle: *Show me what IS unlabeled AND NOT positive.* It turns out that negative samples included in the unlabeled dataset are both unlabeled and not positive. Consequently $G$ addresses this riddle by learning to show the negative samples distribution to $D$.

The Positive Unlabeled (PU) learning problem consists in trying to distinguish positive samples from negative samples by using a PU dataset. We start by detailing the expected PU functionality that we want to incorporate into the GAN discriminator loss function. Let $X \in \mathbb{R}^m$ be the input random variable and $Y \in \{0, 1\}$ its associated label. $X$ can be a positive $X_P$, negative $X_N$ or unlabeled $X_U$ sample which respectively follow the distributions $p_P = p(X|Y = 0)$, $p_N = p(X|Y = 1)$ and $p_U = (1 - \pi_P) \cdot p_N + \pi_P \cdot p_P$, with $\pi_P \in (0, 1)$ the unknown prior which represents the fraction of unlabeled positive samples included in the unlabeled dataset. Let $D : \mathbb{R}^m \to [0, 1]$ be the decision function which is then considered as the discriminator network of the proposed framework. We have $l(\hat{y}, y)$ such that $l : [0, 1] \times [0, 1] \to \mathbb{R}$ is the arbitrary cost function with the predicted output $\hat{y}$ of $D$ for a given sample and the corresponding label $y$ as inputs. $D$ is trained with a PU risk $R_{PU}$ to predict the label value 1 for the unlabeled samples, and the label value 0 for the positive labeled ones. Given the composition of the unlabeled dataset, we develop $R_{PU}(D)$ such that:

$$
\begin{aligned}
R_{PU}(D) &= \mathbb{E}_{x_U \sim \boldsymbol{p_U}}[l(D(x_U), \mathbf{1})] + \mathbb{E}_{x_P \sim \boldsymbol{p_P}}[l(D(x_P), \mathbf{0})] \\
&= (1 - \pi_P) \cdot \mathbb{E}_{x_N \sim \boldsymbol{p_N}}[l(D(x_N), \mathbf{1})] + \pi_P \cdot \mathbb{E}_{x_P \sim \boldsymbol{p_P}}[l(D(x_P), \mathbf{1})] \\
&\quad + \mathbb{E}_{x_P \sim \boldsymbol{p_P}}[l(D(x_P), \mathbf{0})].
\end{aligned}
\tag{1}
$$

We notice that negative samples are associated exclusively to the label value 1 for any $\pi_P$ value. Labeled and unlabeled positive samples follow the same distribution $p_P$. So $p_P$ is associated to both contradictory labels 0 and 1. Consequently, as discussed in annexe section 6.1, we assume it is equivalent to associate the positive distribution $p_P$ with a unified intermediate label value $\delta \in (0,1)$. So if we train $D$ with the risk $R_{PU}$, then $D$ prediction concerning the unlabeled positive samples is shifted away from the label value 1. Thus $D$ predicts the label value 1 exclusively for the negative samples.

Next, let us consider the decision function $D$ as the discriminator of a generative adversarial network. We propose to incorporate the risk $R_{PU}$ into the value function of the original GAN model (Goodfellow et al., 2014) by adding the loss function term $\mathbb{E}_{x_P \sim p_P}[log(1 - D(x_P))]$. Consequently, the training loss function $L_D$ of $D$ is:

$$
\begin{aligned}
L_D(G, D) = \mathbb{E}_{x_U \sim p_U}[logD(x_U)] + \mathbb{E}_{z \sim p_z}[log(1 - D(G(z)))] \\
+ \mathbb{E}_{x_P \sim p_P}[log(1 - D(x_P))],
\end{aligned}
\tag{2}
$$

with $z$ the input random vector of the generative model $G$ such that $G(z)$ is a generated sample. $z$ can follow a uniform or normal distribution. The equation 2 is justified intuitively as follow. Instead of asking to $G$ to converge towards the distribution $p_U$, we want that $G$ learns to converge towards the distribution $p_N$ of negative samples. Thus we propose to replace the standard GAN task *show me what is unlabeled* by the task *show me what is both unlabeled and not positive*. This results in addressing directly the risk $R_{PU}$ to $D$ such that $L_D$ is as follow:

$$
L_D(G, D) = R_{PU}(D) + \mathbb{E}_{z \sim p_z}[l(D(G(z)), 0)],
\tag{3}
$$

with 0 the label associated to the samples $G(z)$ generated by $G$. Let the cost function $l$ used in the proposed D-GAN framework be the binary cross-entropy $H$ such that $l = -H$. So:

$$
\begin{aligned}
L_D(G, D) = \mathbb{E}_{x_U \sim p_U}[-H(D(x_U), 1)] + \mathbb{E}_{x_P \sim p_P}[-H(D(x_P), 0)] \\
+ \mathbb{E}_{z \sim p_z}[-H(D(G(z)), 0)],
\end{aligned}
\tag{4}
$$

We recall that the binary cross-entropy $H$ is defined as below:

$$
H(D(X), Y) = -Y \cdot log(D(X)) - (1 - Y) \cdot log(1 - D(X)),
\tag{5}
$$

with $Y$ the label associated to the $D$ input $X$. Thus $H(D(X), 1) = -log(D(X))$ and $H(D(X), 0) = -log(1 - D(X))$. Consequently, $L_D$ can be developed as in the equation 2. This shows the incorporation of $R_{PU}$ risk (equation 1) inside the D-GAN discriminator loss function (equation 2).

On the other hand, the role of $G$ during the adversarial training is to generate samples considered by $D$ as 1. So, as suggested by Goodfellow et al. (2014) in practice, the training loss function $L_G$ of $G$ is such that $L_G(G, D) = \mathbb{E}_{z \sim p_z}[log(D(G(z)))]$. We recall that exclusively the negative samples are considered as 1 by $D$ thanks to the $R_{PU}$ risk, as shown previously. Consequently this justifies intuitively the $G$ convergence towards the negative samples distribution $p_N$. The corresponding implementation algorithm 1 of this first stage is presented in annexe 6.

Once the D-GAN training is finished, we can start the second step which consists in using a classifier $C$ to distinguish fake generated samples $x_{FN} = G(z)$, which are ideally equivalent to the real negative samples, from real positive labeled samples as illustrated in figure 1. The Divergent-GAN behaviour is only possible if the discriminator $D$ associates the label value 1 exclusively with the negative samples distribution. Nonetheless, in the worst-case scenario where the discriminator $D$ is not able to sufficiently encode the complexity of the boundary between positive and negative samples included in the unlabeled dataset, the D-GAN will behave like the PGAN, which is the best solution in this situation. This is observed experimentally in the section 4.

## 3.2 WITHOUT DISCRIMINATOR BATCH NORMALIZATION

Nowadays, Batch Normalization (BN) (Ioffe & Szegedy, 2015) is considered as a relevant regularization technique commonly used in deep neural networks architectures. Its utility for GANs training

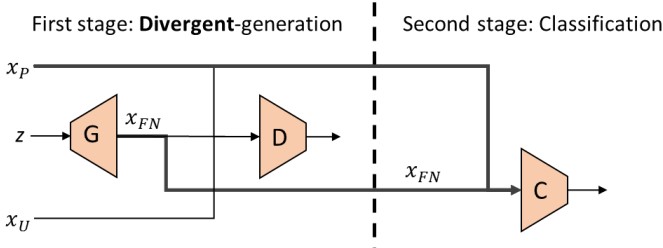

Figure 1: Proposed Positive Unlabeled system: Divergent-GAN Learning model. $x_{FN}$ represents the fake generated samples which are similar to real negative samples $x_N$. $G$ is the generative model. $D$ is the discriminator. $C$ is the classifier used to perform the binary Positive-Negative (PN) classification.

has been highlighted by (Radford et al., 2015) for the DCGAN architecture in order to stabilize the adversarial training. Other variants like the Wasserstein-GAN (Arjovsky et al., 2017) or the Loss-Sensitive GAN (Qi, 2017) confirmed its interest. As developed in (Ioffe & Szegedy, 2015), BN addresses issues like vanishing, exploding gradient problems, and the risk of getting stuck in a poor local minima. BN suggests to reduce the $internal\ covariate\ shift$ problem of the learning model. In this way, a higher learning rate can be used and it can improve significantly the training speed.

But BN regularizes the model, such that a training sample from a given minibatch is considered in conjunction with other examples of this minibatch. This is the consequence of estimating the mean and variance normalization parameters one time per minibatch, and then applying them on each minibatch sample. It turns out that it does not allow to link labeled positive samples to the unlabeled positive samples and to produce a gap between positive and negative samples, when positive samples $x_P$ and unlabeled samples $x_U$ are not in the same training minibatch. To counter this problem, we could imagine to apply BN on a unified minibatch which contains a fraction of each distribution $x_P$, $x_U$ and $x_F$. But the BN effect is influenced a lot by the content of the minibatch on which it is applied. So the fraction $\pi_P$ of positive samples included in $x_U$ will affect the batchnorm effect.

As illustrated experimentally in figure 4, with BN, we observe that the classifier only learns to distinguish minibatches of positive samples from minibatches of unlabeled samples. On the contrary, without the use of BN, both distributions appear in the unlabeled minibatch from the discriminator output point of view. In other words, the negative samples can be distinguished from the positive samples without BN.

However, BN benefits in a more traditional training are not negligible, so we suggest to use another type of normalization which could replace BN role in a PU learning task. Layer Normalization (LN) (Ba et al., 2016), is a frequently used technique with sequential networks as it can be applied for each sequential sample separately. With LN the normalization for a given sample is computed on its resulting output feature map layers and the mean and variance are computed separately for each sample of a minibatch. For this reason, we propose to apply LNs instead of BNs inside our discriminative model $D$. As we can observe in figure 4, it allows the discriminator to keep the behaviour that interests us which is the ability to distinguish unlabeled negative samples to unlabeled positive samples. Additional qualitative experiments in annexe section 6 show that the LN can substitute BN inside the discriminative model structure to generate acceptable images.

The next section presents experimental results which demonstrate the good performance of the proposed approach.

## 4   EXPERIMENTAL RESULTS

This section demonstrates experimentally the D-GAN abilities on simple and complex datasets: We present qualitative results for the counter-example generation, the discriminator normalization effect, and quantitative results for the PU learning image classification task.

Experimental settings are as follow. Minibatches of labeled positive samples, unlabeled samples, and generated samples have the same size. The convolutional classifier $C$ used [2] in the subsection 4.2 is described in the PGAN article (Chiaroni et al., 2018) experiments. $C$ is always trained during 20 epochs. We use the DCGAN (Radford et al., 2015) architecture for all the comparative experiments except for celebA and CIFAR-10 datasets where we respectively adapt the LS-GAN (Qi, 2017) and WGAN-GP (Gulrajani et al., 2017) variants to the D-GAN framework. The D-GAN uses LN instead of BN concerning $D$. Concerning the PU dataset initialization from a standard PN dataset, $\rho$ is the fraction of the positive labeled samples of the initial PN dataset that we unlabel such that they are included into the unlabeled dataset. $\pi_P$ is the fraction that represents these unlabeled positive samples among the unlabeled dataset.

From a qualitative point of view, and on the contrary to the PGAN model, the D-GAN generates samples which are only similar to the counter-examples for diverse data types, as illustrated in figure 3 for a 2D point cloud dataset and in figure 2 for a real images dataset. In order to enable the proposed approach reproducibility, a D-GAN implemented version corresponding to the figure 2 results is available [3] and is applied on the LSGAN model (Qi, 2017). Our code includes also the method implementation, proposed by Chiaroni et al. (2018), to establish a PU training dataset from a fully labeled dataset.

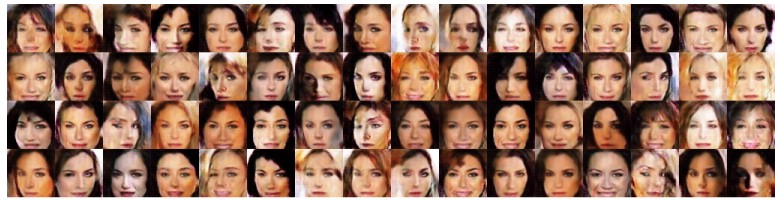

Figure 2: This is a minibatch of fake negative images $x_{FN}$ generated, after 20 epochs iteration, with the D-GAN based on the Loss-Sensitive GAN (Qi, 2017) and by replacing BN with LN. The PU training dataset uses the celebA dataset images with settings $\rho = 0.5$, $\pi_P = 0.3$ and "Male" the positive class chosen arbitrarily. The images size is cropped to 64*64*3. All generated samples are qualitatively relevant fake counter-examples.

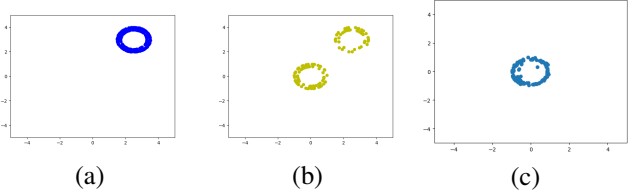

|     (a)     |     (b)     |     (c)     |

Figure 3: D-GAN functioning on clusters of 2D points such that $D$ and $G$ have a multilayer-perceptron structure. (a), (b) and (c) are respectively sets of labeled positive, unlabeled, and generated samples. The generated samples only follow the distribution of the counter-examples.

### 4.1 NORMALIZATION EFFECTS ON THE DISCRIMINATOR

We compare in the figure 4 the ability of $D$ to distinguish the positive from the negative samples distribution when it is trained on a PU dataset with BN, LN layers or without any normalization technique. Smaller is surface intersection between the discriminator output histogram distribution predictions on positive and negative samples included in the unlabeled dataset, better it is.

---

[2] $https://github.com/tensorflow/tensorflow/blob/master/$
$tensorflow/examples/tutorials/mnist/mnist\_softmax.py$
[3] after the anonymous reviewing period

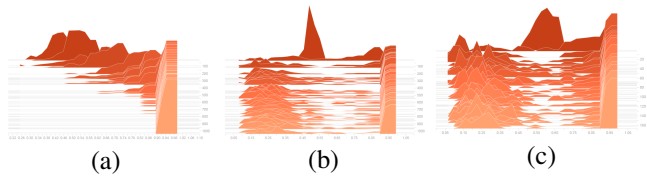

(a)       (b)       (c)

Figure 4: Normalization effect on the histogram distribution of output values predicted by $D$ on an unlabeled samples minibatch. (a) is with BN, (b) without any normalization, and (c) with LN. The axe in depth represents the training iterations. The lighter histogram corresponds to the last training iteration. Settings are with positive class 8 and negative class 3 of MNIST dataset, with $\rho = 0.5$ and $\pi_P = 0.3$. We observe that BN in the figure 4(a) does not enable to distinguish positive from negative samples when the discriminator is trained on a PU dataset. On the contrary, LN (figure 4(c)) enables it for the reasons specified in the subsection 3.2. Thus with LN in $D$, $G$ can converge exclusively towards the unlabeled negative samples distribution $p_N$, as expected.

## 4.2 DIVERGENT-GAN FOR POSITIVE UNLABELED LEARNING

We compare the D-GAN accuracy prediction to GenPU on the table 2 for the simple One vs. One task with few positive labeled samples. The GenPU method presents better results for fewer labeled positive samples. The D-GAN presents Accuracy performances better than the GenPU method for one hundred of positive labeled samples after 40 epochs training iterations for the D-GAN counter-examples generation step. Moreover, it is interesting to observe that the proposed approach globally slightly outperforms nnPU and uPU methods without using prior knowledge.

Table 2: Comparative results of Accuracy prediction performances on MNIST in a One vs. One mode. $N_P$ is the number of positive labeled samples, and $N_U$ is the number of unlabeled samples which mixes all the rest of positive and negative samples. GenPU, uPU and nnPU results come from (Hou et al., 2018).

| One vs. One | '3'vs.'5' | | | | '8'vs.'3' | | | |
|---|---|---|---|---|---|---|---|---|
| Dataset: MNIST | D-GAN (proposed) | GenPU | nnPU | uPU | D-GAN (proposed) | GenPU | nnPU | uPU |
| No need of prior knowledge | ✓ | × | × | × | ✓ | × | × | × |
| $N_P$=100 : $N_U$=9900 | **0.987** | 0.983 | 0.969 | 0.914 | **0.989** | 0.982 | 0.974 | 0.932 |
| $N_P$=50 : $N_U$=9950 | 0.964 | **0.982** | 0.966 | 0.854 | 0.974 | **0.979** | 0.965 | 0.873 |

Concerning the relatively more challenging One vs. Rest task, the D-GAN globally outperforms the PGAN and RP methods in terms of F1-Score predictions performances on the table 3 both on MNIST and CIFAR-10 datasets.

Moreover, the D-GAN demonstrates a steadier behaviour than PGAN against the overfitting problem as illustrated in figure 5(b).

Table 3: Comparative results of two-stage PU methods not using prior knowledge. These are the average F1-Score prediction performances in a One vs. Rest mode on MNIST and CIFAR-10 with respectively 35 Divergent-GAN epochs on MNIST and 275 Divergent-WGAN-GP epochs on CIFAR-10. PGAN and RP results come from (Chiaroni et al., 2018). PNGAN represents a PN (PU when $\pi_P = 0$) training such that the negative samples are replaced by WGAN generated ones. This highlights the GANs data augmentation effect on complex datasets like CIFAR-10.

| One vs. Rest | ref | | $\rho = 0.5, \pi_P = 0.1$ | | | $\rho = 0.5, \pi_P = 0.3$ | | | $\rho = 0.5, \pi_P = 0.5$ | | | $\rho = 0.5, \pi_P = 0.7$ | | |
|---|---|---|---|---|---|---|---|---|---|---|---|---|---|---|
| Datasets | PN | PNGAN | D-GAN | PGAN | RP | D-GAN | PGAN | RP | D-GAN | PGAN | RP | D-GAN | PGAN | RP |
| $AVG_{\text{MNIST}}$ | 0.993 | 0.988 | **0.99** | 0.965 | 0.967 | **0.984** | 0.958 | 0.975 | **0.972** | 0.946 | 0.951 | 0.922 | 0.875 | **0.933** |
| $AVG_{\text{CIFAR-10}}$ | 0.680 | 0.812 | **0.749** | 0.745 | 0.622 | 0.727 | **0.760** | 0.730 | **0.749** | 0.748 | 0.716 | **0.713** | 0.702 | 0.684 |

In figure 5(a), without BN or LN, the D-GAN method gets a faster and better $Accuracy$ prediction performance than the PGAN when both are trained under the same conditions. In figure 5(b), the D-GAN with LN follows the learning training rythm of the PGAN with BN. The D-GAN also

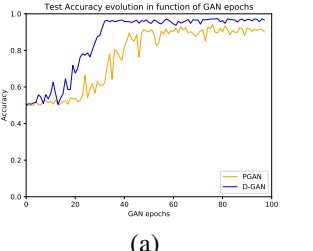 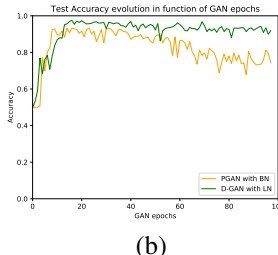

(a)         (b)

Figure 5: Classifier test Accuracy evolution in function of the GAN epochs (first stage). 8 vs. Rest MNIST task, with $\rho = 0.5$ and $\pi_p = 0.5$. (a) D-GAN and PGAN are trained without normalization layers. (b) D-GAN and PGAN are respectively trained with LN and BN inside the discriminator.

has a steadier behaviour once the Accuracy progression is finished: it reduces the PGAN first step overfitting problem.

## 5 CONCLUSION

To conclude, we have incorporated into the discriminator $D$ loss function a standard PU risk to address the PU learning challenge. In this way, the proposed model generates relevant counter-examples from a PU dataset without prior knowledge. It outperforms two-stage state of the art PU learning performances by addressing previous GAN-based PU methods issues: -It reduces the overfitting PGAN first stage problem. -It addresses the GenPU mode collapse issue by keeping the practical standard GANs architecture such that it is easily adaptable to recent GANs variants. A side contribution of this article is to identify a discriminator batch-normalization critical issue appearing when the discriminator manipulates multiple minibatches distributions.

We strongly believe that the proposed approach stability and prediction performances still have the potential to be improved by taking the best of the representation learning and PU learning domains. For example, recent promising GAN training approaches (Karras et al., 2018) not mandatorily using BN should be suitable to extend the proposed approach for large-scaled image datasets. In addition, premices of Noisy labeled learning results presented in annexe section 6.3 propose to extend the presented approach to a unified generative framework which could output relevant generated labeled samples even from a weak, noisy labeled or incremental dataset with varying fractions of unlabeled positive samples.

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

# 6 ANNEXE

The algorithm 1 is the implementation of the proposed approach. It includes the option for the noisy labeled learning task.

The table 4 details for each class the average F1-Score results presented in Table 3.

The figure 9 presents the same minibatch generated as in the figure 2, but in a larger size.

## 6.1 AN EXPERIMENTAL ANALYSIS OF THE RISK $R_{PU}$

If $D$ does not overfit labeled and unlabeled positive samples, let $\delta \in (0, 1)$ be the label associated to all positive samples such that we propose the Positive Negative risk $R_{PN}(D, \delta) = R_{PU}(D)$. $\delta$ represents the intermediate label value equivalent of associating, in the risk $R_{PU}$, both contradictory labels 0 and 1 respectively to labeled and unlabeled positive samples. We develop the risk $R_{PN}$ as follow:

$$R_{PN}(D, \delta) = (1 - \pi_P) \cdot \mathbb{E}_{x_N \sim p_N}[l(D(x_N), 1)] + (1 + \pi_P) \cdot \mathbb{E}_{x_P \sim p_P}[l(D(x_P), \delta)]. \tag{6}$$

As illustrated in figure 6, in practice we can train $D$ with the risk $R_{PU}$ to output a value $\hat{y}$ close to $\delta$ when the input is a positive sample, and close to 1 when the input is a negative sample. We can observe that the global minimum of the approximated risk $\hat{R}_{PN}(D, \delta)$ corresponds to the global maximum of the distribution predicted by $D$ in output for positive unlabeled samples. Moreover, the distribution of the negative samples is centered on the label associated to the unlabeled samples during the training, which coincides also with the equation 6 formulation. This illustrates experimentally that $D$ predicts the label value 1 exclusively for the negative samples.

## 6.2 NEEDING PRIOR KNOWLEDGE MAY NOT BE SUITABLE FOR INCREMENTAL APPLICATIONS

In the context of real learning applications like incremental learning (Craye et al., 2018), (Hadsell et al., 2009), the fraction $\pi_P$ of unlabeled positive samples can vary continuously at each new incremental unlabeled minibatch of samples to process. However, existing PU learning methods using prior knowledge, like uPU and nnPU implicitly do the assumption that the prior knowledge holds the same during all the training. A PU learning method which does not need prior knowledge as the D-GAN proposed approach could be extended to such real applications.

## 6.3 AN EXTENSION TO NOISY LABELED LEARNING AND FEW LABELED DATA RESSOURCES

If a generative model in an adversarial training only converges towards the subdivision of a distribution which is considered as the closest to a given label from the discriminator point of view, then we can extend the proposed approach to the noisy labeled learning task. More precisely, the D-GAN architecture can be modified to include a generative model $G_P$ trained to generate fake positive samples. This is achieved by adding the loss function term $\mathbb{E}_{z \sim p_z}[log(1 - D(G_P(z)))]$ in the minimax value function. Furthermore, this extension also allows to manage problems where we only have few labeled positive samples. The positive generator $G_P$ will converge towards the distribution

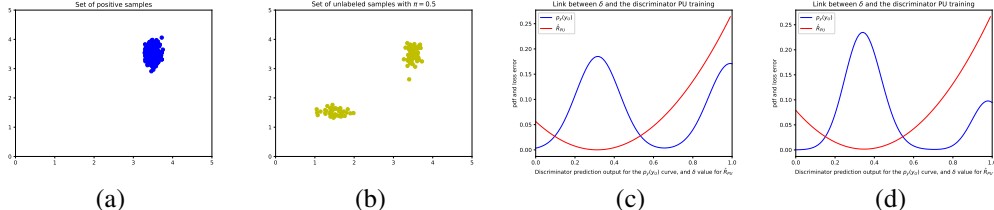

(a)   (b)   (c)   (d)

Figure 6: Link between the PN loss function suggested (equation 6) and the distribution of the discriminator output predictions for an unlabeled minibatch. For this experiement, $l$ is the mean squared error (MSE) and $D$ is a multi-layer perceptron. $D$ has been trained to distinguish a 2D gaussian distribution to another one by using the risk $R_{PU}$ on a PU dataset. (a) Shows a set of 2D points considered as positive samples. (b) Shows a set of 2D points considered as unlabeled samples. Both curves in (c) and (d) have been normalized to get a better visualization. For (c), $p_Y(D(x_U))$ (in blue) represents the probability distribution of $D$ predicted outputs for a minibatch of unlabeled samples, with $\pi_P = 0.5$. $\hat{R}_{PN}(D, \delta)$ (in red) represents the PN risk computed in function of $\delta$ with the equation 6 on a minibatch of positive and negative labeled samples, once $D$ is trained with $R_{PU}$ risk. (d) shows the same curves as in (c) but by giving in input a concatenation of an unlabeled minibatch with a positive labeled minibatch. Unlabeled and labeled positive samples provide a unified prediction output distribution.

considered as the most positive, the closest to zero in our case, which includes also the unlabeled positive samples perceived by the discriminator at the same location that the labeled ones as illustrated in figure 6 (d). We argue that it is indirectly the same phenomenon that allows the GenPU (Hou et al., 2018) to get relatively good accuracy predictions with few labeled positive samples. Because the discriminator task becomes more difficult, the weight $\alpha$ is applied on the unlabeled loss function $\mathbb{E}_{x_U \sim p_U}[log(D(x_U))]$ and on the positive one $\mathbb{E}_{x_P \sim p_P}[log(1 - D(x_P))]$. It ensures the guidelines that both generators $G$ and $G_P$ have to follow. The boolean variable $noisyLabelsMode$ of the algorithm 1 is set to $True$ to activate this functionality.

This noisy labeled learning D-GAN proposed extension cannot capture the entire positive samples distribution. Nonetheless, this extension presents an interesting behaviour as illustrated in the figure 7 showing the results obtained on clusters of 2D point clouds with the D-GAN, when its noisy labeled mode is activated.

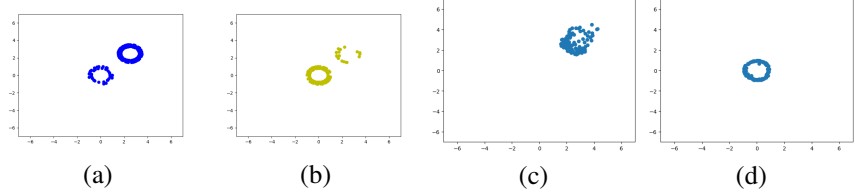

(a)   (b)   (c)   (d)

Figure 7: D-GAN Noisy-labeled version functioning on sets of 2D points such that (a), (b), (c) and (d) are respectively sets of real noisy labeled positive (20 percents of noise), real noisy labeled negative (20 percents of noise), fake generated positive and fake generated negative samples. $D$, $G$ and $G_P$ have a multilayer-perceptron structure. The generated samples only follow the ditribution of the well-labeled samples of the input sets.

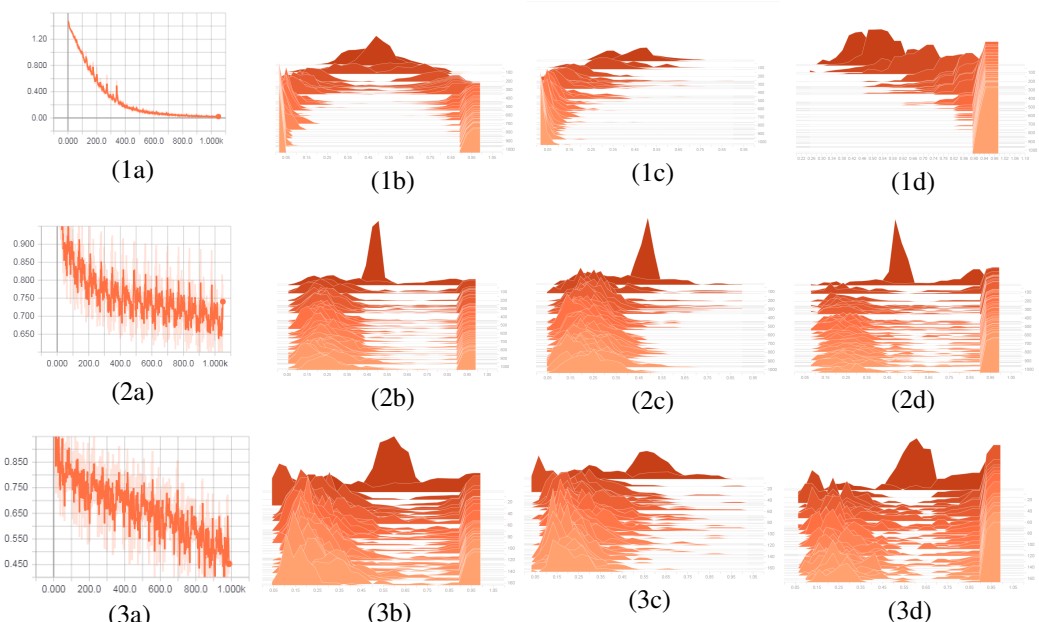

Figure 8: Details of figure 4 Batch Normalization effect on discriminator output predictions on distinct Positive and Unlabeled minibatches. Settings are with positive class 8 and negative class 3 of MNIST dataset, with $\rho = 0.5$ and $\pi_P = 0.3$. The results on the first row are done with BN, without on the second row, and with LN on the third row. The first column (a) is the mean $D$ loss value estimated in function of training iterations. The second column (b) is the histogram distributions of output values predicted by $D$ on the positive and unlabeled minibatches. The third column (c) is the histogram distribution of output values predicted by $D$ on the positive minibatch. The fourth column (d) is the histogram distribution of output values predicted by $D$ on the unlabeled minibatch. The axe in depth represents the training iteration axe on which the clearest histograms gradually correspond to the closest training iterations. Other alternative normalization techniques have been applied to GANs, such as weight normalization (Karras et al., 2018) or the Virtual Batch Normalization (VBN) (Salimans et al., 2016) where they essentially mention the same problem for BN. However, VBN parameters strongly depend to their "reference batch" composition. This is a severe limitation if the fraction of unlabeled positive samples changes during the training, as it can happen for example in a real incremental learning application. Moreover, VBN in (Salimans et al., 2016) is only used in the generator, which is not what we are looking for. For PU learning task, BN is not a problem inside the generator structure.

---

**Algorithm 1** Minibatch stochastic gradient descent training of the Divergent-GAN

---

Initialize $noisyLabelsMode = false$.
Initialize $\alpha = 2$.
**for** number of training iterations **do**
    **for** $k$ steps **do**
        Sample minibatch of $m$ noise samples $\{z^{(1)}, ..., z^{(m)}\}$ from noise prior $p_G(z)$.
        Sample minibatch of $m$ unlabeled examples $\{x_U^{(1)}, ..., x_U^{(m)}\}$ from data distribution $p_U$.
        Sample minibatch of $m$ positive labeled examples $\{x_P^{(1)}, ..., x_P^{(m)}\}$ from data distribution $p_P$.

        Update the discriminator $D$ by descending its stochastic gradient:
        **if** $noisyLabelsMode$ is $false$ **then**

$$\nabla_{\theta_D} \frac{1}{m} \sum_{i=0}^{m} logD(x_U^{(i)}) + log[1 - D(G(z^{(i)}))] + log[1 - D(x_p^{(i)})]$$

        **else**

$$\nabla_{\theta_D} \frac{1}{m} \sum_{i=0}^{m} \alpha \cdot \left( logD(x_U^{(i)}) + log[1 - D(x_p^{(i)})] \right) + log[1 - D(G(z^{(i)}))]$$

$$+logD(G_P(z^{(i)}))$$

        **end if**
    **end for**
    Sample minibatch of $m$ noise samples $\{z^{(1)}, ..., z^{(m)}\}$ from noise prior $p_G(z)$.
    Update the generator $G$ by descending its stochastic gradient:

$$\nabla_{\theta_G} \frac{1}{m} \sum_{i=0}^{m} log[D(G(z^{(i)}))]$$

    **if** $noisyLabelsMode$ is $true$ **then**
        Update the generator $G_P$ by descending its stochastic gradient:

$$\nabla_{\theta_{G_P}} \frac{1}{m} \sum_{i=0}^{m} log[1 - D(G_P(z^{(i)}))]$$

    **end if**
**end for**
The gradient-based updates can use any standard gradient-based learning rule. We use Adam in our experiments.

---

Table 4: Comparative results of F1-Score prediction performances on MNIST and CIFAR-10 in a One vs. Rest mode with 20 classifier epochs.

| One vs. Rest | ref | | $\rho = 0.5, \pi_P = 0.1$ | | | $\rho = 0.5, \pi_P = 0.3$ | | | $\rho = 0.5, \pi_P = 0.5$ | | | $\rho = 0.5, \pi_P = 0.7$ | | |
|---|---|---|---|---|---|---|---|---|---|---|---|---|---|---|
| Datasets | PN | PNGAN | D-GAN | PGAN | RP | D-GAN | PGAN | RP | D-GAN | PGAN | RP | D-GAN | PGAN | RP |
| 0 | 0.997 | 0.992 | **0.996** | 0.974 | 0.992 | **0.994** | 0.973 | 0.955 | 0.984 | 0.973 | **0.991** | **0.965** | 0.902 | 0.88 |
| 1 | 0.998 | 0.993 | **0.995** | 0.971 | **0.995** | 0.994 | 0.979 | **0.996** | 0.988 | 0.958 | **0.994** | 0.952 | 0.863 | **0.993** |
| 2 | 0.99 | 0.987 | **0.992** | 0.972 | 0.975 | **0.99** | 0.959 | 0.923 | **0.98** | 0.947 | 0.936 | 0.892 | 0.914 | **0.987** |
| 3 | 0.996 | 0.996 | **0.993** | 0.963 | 0.991 | 0.97 | 0.953 | **0.991** | **0.957** | 0.934 | 0.882 | **0.914** | 0.885 | 0.829 |
| 4 | 0.997 | 0.992 | **0.989** | 0.964 | 0.972 | 0.989 | 0.952 | **0.995** | **0.972** | 0.945 | 0.933 | 0.923 | 0.914 | **0.977** |
| 5 | 0.993 | 0.981 | **0.988** | 0.974 | 0.985 | **0.976** | 0.95 | 0.943 | **0.969** | 0.949 | 0.91 | 0.894 | 0.873 | **0.973** |
| 6 | 0.992 | 0.989 | **0.992** | 0.962 | 0.928 | 0.99 | 0.959 | **0.992** | 0.986 | 0.971 | **0.993** | 0.961 | 0.944 | **0.99** |
| 7 | 0.995 | 0.99 | **0.982** | 0.962 | 0.947 | 0.983 | 0.96 | **0.991** | 0.969 | 0.926 | **0.988** | 0.918 | 0.737 | **0.979** |
| 8 | 0.995 | 0.985 | **0.985** | 0.949 | 0.929 | 0.981 | 0.941 | **0.982** | **0.966** | 0.922 | 0.941 | **0.88** | 0.849 | 0.818 |
| 9 | 0.981 | 0.977 | **0.986** | 0.959 | 0.954 | 0.972 | 0.956 | **0.979** | **0.947** | 0.939 | 0.941 | **0.921** | 0.865 | 0.904 |
| $AVG_{MNIST}$ | 0.993 | 0.988 | **0.99** | 0.965 | 0.967 | **0.984** | 0.958 | 0.975 | **0.972** | 0.946 | 0.951 | 0.922 | 0.875 | **0.933** |
| Plane | 0.727 | 0.803 | 0.759 | **0.818** | 0.669 | 0.749 | 0.784 | **0.795** | 0.736 | **0.758** | 0.743 | 0.715 | **0.731** | 0.718 |
| Auto | 0.78 | 0.865 | 0.766 | **0.801** | 0.695 | 0.749 | 0.737 | **0.829** | 0.793 | 0.789 | **0.798** | 0.757 | 0.734 | **0.783** |
| Bird | 0.447 | 0.775 | 0.686 | **0.688** | 0.56 | 0.706 | **0.744** | 0.68 | 0.692 | **0.694** | 0.644 | 0.68 | **0.688** | 0.542 |
| Cat | 0.5 | 0.698 | **0.695** | 0.658 | 0.384 | 0.713 | **0.722** | 0.651 | 0.707 | **0.718** | 0.67 | **0.712** | 0.69 | 0.698 |
| Deer | 0.698 | 0.8 | **0.768** | 0.68 | 0.605 | 0.681 | **0.708** | **0.708** | 0.72 | 0.708 | 0.64 | **0.672** | 0.633 | 0.602 |
| Dog | 0.567 | 0.805 | **0.716** | 0.632 | 0.539 | 0.68 | **0.756** | 0.648 | 0.74 | **0.746** | 0.733 | 0.704 | 0.678 | **0.712** |
| Frog | 0.691 | 0.812 | 0.758 | **0.837** | 0.666 | 0.786 | 0.793 | **0.794** | 0.78 | **0.788** | 0.769 | 0.723 | **0.75** | 0.749 |
| Horse | 0.786 | 0.845 | **0.697** | 0.693 | 0.653 | 0.685 | **0.757** | 0.723 | 0.74 | 0.751 | **0.759** | **0.713** | 0.675 | 0.711 |
| Ship | 0.832 | 0.879 | **0.851** | 0.821 | 0.764 | 0.784 | 0.809 | **0.831** | **0.797** | 0.775 | 0.785 | 0.728 | 0.716 | **0.755** |
| Truck | 0.771 | 0.839 | 0.794 | **0.822** | 0.685 | 0.741 | **0.786** | 0.637 | **0.778** | 0.754 | 0.617 | **0.732** | 0.724 | 0.564 |
| $AVG_{CIFAR-10}$ | 0.680 | 0.812 | **0.749** | 0.745 | 0.622 | 0.727 | **0.760** | 0.730 | **0.749** | 0.748 | 0.716 | **0.713** | 0.702 | 0.684 |

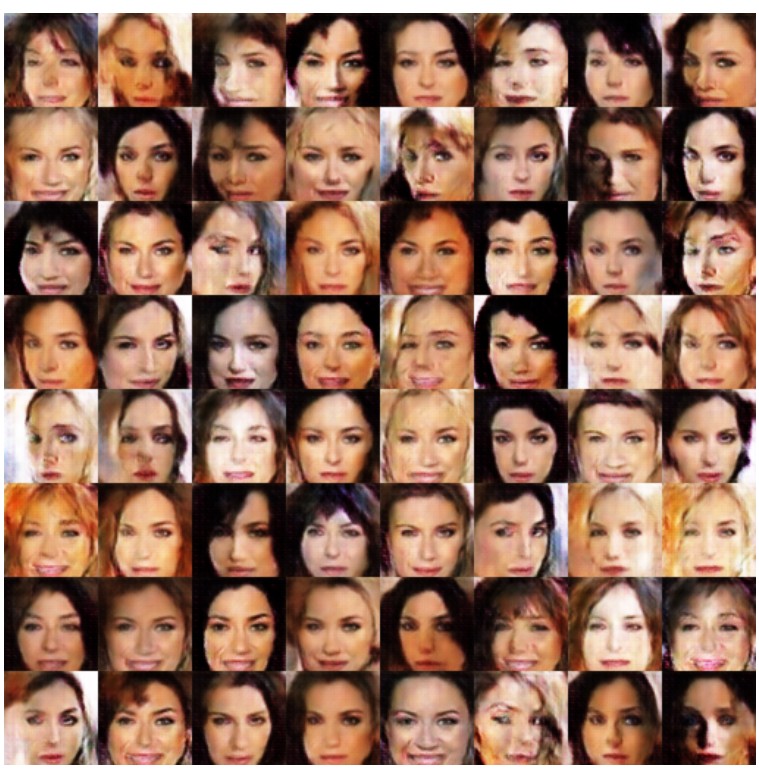

Figure 9: Minibatch of counter-examples (women faces) generated after 20 epochs with the proposed D-GAN framework adapted to the LSGAN-GP variant. The D-GAN is applied on a PU celebA dataset (celebrity RGB images of size 64*64), with 30% of positive samples (men faces) among the unlabeled ones. We can visually observe that 100% of the generated samples follow the counter-examples (women faces) distribution. The LSGAN-GP code is available at: $https://github.com/maple-research-lab/lsgan-gp-alt$.

