# OpenReview forum: "D-GAN: Divergent generative adversarial network for positive unlabeled learning and counter-examples generation"
_ICLR.cc/2019/Conference_

### Official Review · AnonReviewer2 · 2018-11-02
**Too many issues**

**Rating:** 3
**Confidence:** 5

**Review:**

This paper proposed another GAN-based PU learning method. The mathematics in this paper is not easy to follow, and there are many other critical issues.

*****

The clarity is really an issue. First of all, I cannot easily follow the meanings behind the equations. I guess the authors first came up with some concrete implementation and then formalize it into an algorithm. Given the current version of the paper, I am not sure whether this clarity of equations can be fixed without an additional round of review or not.

Moreover, the logic in the story line is unclear to me, especially the 3rd paragraph that seems to be mostly important in the introduction. There are two different binary classification problems, of separating the positive and negative classes, and of separating the given and generated data. I cannot see why the generated data can serve as negative data. This paragraph is discussing GenPU, PGAN and the proposed method, and consequently the motivation of the current paper does not make sense at least to me.

*****

The paper classified PU learning methods into two categories, one-stage methods and two-stage methods. This is interesting. However, before that, they should be classified into two categories, for censoring PU learning and for case-control PU learning. The former problem setting was proposed very early and formalized in "learning classifiers from only positive and unlabeled data", KDD 2008; the latter problem setting was proposed in "presence-only data and the EM algorithm", Biometrics 2009 and formalized in "analysis of learning from positive and unlabeled data", NIPS 2014. Surprisingly, none of these 3 papers was cited. By definition, GAN-based PU learning belongs to the latter problem setting while Rank Prune can only be applied to the former but was included as a baseline method.

The huge difference between these two settings and their connections to learning with noisy labels are known for long time. To be short, class-conditional noise model corrupts P(Y|X) and covers censoring PU, mutual contamination distribution framework corrupts P(X|Y) and covers case-control PU, and mathematically mutual contamination distribution framework is more general than class-conditional noise model and so is case-control PU than censoring PU. See "learning from corrupted binary labels via class-probability estimation", ICML 2015 for more information where the above theoretical result has been proven. An arXiv paper entitled "on the minimal supervision for training any binary classifier from only unlabeled data" has some experimental results showing that methods for class-conditional noise model cannot handle mutual contamination distributions. The situation is similar when applying censoring PU methods to case-control PU problem setting.

Furthermore, the class-prior probability pi is well-defined and easy to estimate in censoring PU, see "learning classifiers from only positive and unlabeled data" mentioned above. However, it is not well-defined in case-control PU due to an identifiability issue described in "presence-only data and the EM algorithm" mentioned above. Thus, the target to be estimated is defined as the maximal theta such that theta*P(X|Y)<=P(X) following "estimating the class prior and posterior from noisy positives and unlabeled data", NIPS 2016. BTW, "mixture proportion estimation via kernel embedding of distributions" is SOTA in class-prior estimation; the previous NIPS paper was written earlier and accepted later.

In summary, as claimed in the paper and shown in Table 1 in the introduction, all discriminative PU methods and GenPU require to know pi for learning. This is true, but this is because they are designed for a more difficult problem setting---learning classifiers and estimating pi are both more difficult. Lacking some basic knowledge of PU learning is another big issue.

*****

The novelty is to be honest incremental and thus below the bar of ICLR. The significance is similarly poor, due to that the experiments mixed up methods for censoring PU and those for case-control PU. What is more, F1-score is a performance measure for information retrieval rather than binary classification. We all know GANs are pretty good at MNIST but not CIFAR-10. In fact, GenPU has a critical issue of mode collapse, and this is why GenPU reports 1-vs-1 rather than 5-vs-5 on MNIST. Even though, I still think GenPU makes much more sense than PGAN and D-GAN.

---

> ### Author Response · Authors · 2018-11-25
> **Answers (Part 1)**
>
> Thanks for your constructive review,
> Your comments indicate that the text and equations are not clear enough and that some previous state of the art methods were omitted. We understand that the lack of clarity can be an issue. We made during this rebuttal period a clarification effort.
> Moreover, please find as follow the answers to your comments.
>
>
> *****
>
>
> “I cannot easily follow the meanings behind the equations.”
>
> We have clarified the equations.
>
>
> “I cannot see why the generated data can serve as negative data.” “This paragraph is discussing GenPU, PGAN and the proposed method, and consequently the motivation of the current paper does not make sense at least to me.”
>
> GANs are known to be relevant because of their ability of finding a boundary between real and generated samples: A GAN discriminator is trained to find autonomously the best metric to evaluate the generated samples quality. This metric is considered as more relevant than previous ones such as the auto-encoders per-pixel reconstruction loss function.
> The GAN-based PU approaches main idea is to exploit this GAN benefit to address a PU learning problem: The initial goal of GANs is to imitate the unlabeled distribution. In the context of the PU task, this goal is adapted to identify and imitate autonomously the distribution of relevant counter-examples hidden in the unlabeled dataset.
>
> The motivation in this paragraph is to discuss the previous GAN-based approaches following issues:
> -	GenPU issues: GenPU is not easily adaptable to the current GAN state of the art (fast) evolutions because of its untraditional adversarial framework. Moreover, GenPU uses prior knowledge. This is unpractical for example on some real application incremental datasets in which the fraction pi value can change continuously at each new training minibatch.
> -	PGAN issue: It has a first stage overfitting problem when it is applied on relatively simple datasets as MNIST. In fact, it is mentioned in their article: “It is also known that a GAN is not perfect in its operation when it is applied to high dimensional data, … Thus it is possible to estimate the non-zero distance d computed into the cost function of Db”. In other words, the PGAN exploits the GANs convergence defaults to address the PU learning problem.
>
> The proposed approach overcomes the above enumerated issues while keeping their respective advantages. This is done by using a different technique: The D-GAN directly incorporates a PU learning risk into the discriminator loss function. This guides naturally the generator to converge towards the distribution of the negative samples included in the unlabeled dataset.
>
>
> *****
>
>
> “The paper classified PU learning methods into two categories, one-stage methods and two-stage methods. This is interesting. However, before that, they should be classified into two categories, for censoring PU learning and for case-control PU learning.”
>
> Previous relevant state of the art articles, like nnPU, classify PU learning methods in two-stage and one-stage categories. The article nnPU (“Positive-Unlabeled Learning with Non-Negative Risk Estimator”, NIPS 2017) says: “Existing PU methods can be divided into two categories based on how U data is handled. The first category (e.g., [11, 12]) identifies possible negative (N) data in U data, and then performs ordinary supervised (PN) learning; the second (e.g., [13, 14]) regards U data as N data with smaller weights.”.
>
> GAN-based approaches generate samples in the first step, and they perform ordinary PN learning during the second step by considering the generated samples as relevant counter-examples. RP prepares a PN dataset from a PU dataset. Thus it is relevant to classify into the same category (two-stage) RP, and GAN-based approaches (D-GAN, PGAN, GenPU).
>
> We introduce these categories (one-stage/two-stage) because our goal is to focus the attention on methods which aim at producing a relevant PN dataset from a PU dataset.

---

> > ### Author Response · Authors · 2018-11-25
> > **Answers (Part 2)**
> >
> > Concerning the state of the art,
> >
> >  “none of these 3 papers was cited”
> >
> > We thank you for quoting all these relevant articles. We agree with you that it is interesting to keep in mind the founders articles. We focused the attention in our article on the most recent PU learning approaches which achieve the state of the art prediction performances. The link with the quoted articles is as follow:
> > -	concerning the censoring PU learning, RP can be considered as an improvement of Elk08 ("learning classifiers from only positive and unlabeled data", KDD 2008) method as mentioned in their article: “Rank Pruning leverages Elk08 …”;
> > -	concerning the Case-control PU learning, nnPU (“Positive-Unlabeled Learning with Non-Negative Risk Estimator”; NIPS 2017) addresses the overfitting problem of uPU (“Convex formulation for learning from positive and unlabeled data”; ICML 2015). uPU is an improvement of uPU-2014 (“analysis of learning from positive and unlabeled data”; NIPS 2014), such that uPU-2015 proposes a convex formulation by using “different loss functions for positive and unlabeled samples”. This improvement reduces the computational cost of uPU-2014 method.
> >
> >  We will add the three relevant citations that you quoted to the introduction.
> >
> >
> > “By definition, GAN-based PU learning belongs to the latter problem setting while Rank Prune can only be applied to the former but was included as a baseline method.”
> >
> > Rank Prune (RP) addresses standard PU learning problem as highlighted in the table 2 of their article.
> > RP is a baseline method in the context of the presented D-GAN method because:
> > -	both are two-stage methods such that during the first stage they prepare a PN dataset for the second stage (classifier step);
> > -	both (RP and D-GAN) address the PU learning problem without the need of prior knowledge. What is more, RP achieves state of the art performances as presented in their article;
> > -	both follow a similar reasoning such that:
> >       o	the RP method consists in selecting the confident examples during the first step;
> >       o	the generator G of the D-GAN method consists in learning the distribution of the samples considered as the closest to the value “y=1” (label associated to the unlabeled negative samples) by the discriminator D;
> > So both methods exploit exclusively the samples predicted with the higher confidence.
> >
> > This clarifies why RP can be considered as a baseline in the context of this study. D-GAN method has more points in common with RP than with nnPU method.
> >
> > Our aim is not to center the discriminator predictions for positive samples on the corresponding label value (y=0 in our case).
> > In our case, we train G to generate samples considered by D as “y=1”. Thus we care about two things, which can be obtained for any fraction pi between 0 and 1, as follow:
> > -	the fact that unlabeled positive samples are considered by D as distant to the label value “y=1”, thanks to the fact that they follow the same distribution as the positive labeled samples which are associated during the training to the contradictory label “y=0”. Consequently, in practice D predicts an intermediate value between “y=1” and “y=0” for the positive samples distribution;
> > -	the fact that negative samples included in the unlabeled dataset are exclusively associated by D to their expected label “y=1”;
> >
> > In this way, negative samples are considered as the most “y=1” by D. This enables G convergence towards “y=1” samples: the negative samples.
> >
> >
> > “all discriminative PU methods and GenPU require to know pi for learning.”
> >
> > If we suppose that the D-GAN method is a discriminative method, then the presented results (tables 2, 3, 4, and figures 3, 4, 6) demonstrate experimentally that the discriminative methods do not necessarily need prior knowledge to achieve state of the art predictions, as this is the case for the D-GAN.
> >
> > Thus from this point of view, we can consider that the D-GAN proposed approach is a novel interesting discriminative approach not using pi for learning the counter-examples distribution.
> >
> > The intuition behind the D-GAN is based on an obvious practical phenomenon: With the PU risk proposed (equation 1), the negative samples are always considered as “1” for any fraction pi value.
> >
> >
> > *****

---

> > > ### Author Response · Authors · 2018-11-25
> > > **Answers (Part 3)**
> > >
> > >
> > > *****
> > >
> > >
> > > “The novelty is to be honest incremental”
> > >
> > > The D-GAN addresses the task to generate relevant counter-examples from a PU dataset in a different way than the previous GAN-based PU learning approaches.
> > >
> > > -	GenPU convergence is inspired by the original GAN convergence presented in (“Generative adversarial nets”; 2014 NIPS). The GenPU main idea is in this GenPU article sentence: “Du is aimed at separating the unlabelled training samples from the fake samples of both Gp and Gn”. That enables the global system GenPU convergence such that “pi Pgp + (1-pi) Pgn -> Pu”, with Pgp the distribution of positive samples generated by Gp, Pgn the distribution of the negative samples generated by Gn, and Pu the distribution of unlabeled samples (real ones). However, the same reasoning can be expressed using one single generator Gn if we replace the generated positive samples by the positive labeled samples that we have in a PU dataset. Thus training five different models to address standard PU learning challenge is not necessary. This reasoning is different to the D-GAN one.
> > >
> > > -	PGAN is trained to converge towards the unlabeled dataset distribution during the first step. The PGAN exploits GANs imperfections such that the generated distribution at the adversarial equilibrium is still separable from the unlabeled samples distribution by a classifier. The PGAN method does not focus the generator G convergence towards the counter-examples distribution. The proposed D-GAN approach enables the G convergence exclusively towards the latter.
> > >
> > > Main contribution: We propose to incorporate a PU risk inside the discriminator loss function.
> > > We show that a GAN can solve by itself a Positive Unlabeled learning task if the problem is well formulated: We combine the risk Rpu with the discriminator GAN loss function. That enables the G convergence to the distribution of counter-examples included in the unlabeled dataset.
> > >
> > > The side contribution: highlight of a Batch Normalization (BN) (Ioffe & Szegedy, 2015) critical issue.
> > >  A learning model manipulating different training minibatches distributions should not use BN. Alternative normalization techniques are discussed and tested in the context of the proposed framework which manages positive, unlabeled and generated training minibatches.
> > >
> > > The both enumerated contributions are presented in the article. In addition, the former one presents a thinking different to the previous GAN-based PU approaches.
> > >
> > >
> > > “The significance is similarly poor, due to that the experiments mixed up methods for censoring PU and those for case-control PU.”
> > >
> > > D-GAN is compared to GenPU, nnPU, PGAN, and RP methods achieving state of the art prediction performances. More is better than less.
> > >
> > >
> > > “F1-score is a performance measure for information retrieval rather than binary classification.”
> > >
> > > F1-Score metric is relevant in the context of the One vs. Rest challenge presented because:
> > > -	This metric is used in the RP and PGAN articles reported results. Thus we used the same metric to compare our results to the PGAN article reported results.
> > > -	The F1-Score evaluates the ability of a binary classifier to predict correctly the positive samples. One vs. Rest challenge focuses the attention on the examples of our class of interest: the positive class.
> > > Accuracy metric has been also used in table 2 in the context of the One vs. One task, and in figure 6 to evaluate the overfitting problem.
> > >
> > > Nonetheless, we take into consideration this comment.
> > >
> > >
> > > “We all know GANs are pretty good at MNIST but not CIFAR-10.”
> > >
> > > Thanks for highlighting this point.
> > > Another interest of our article is that we demonstrate experimentally the contrary on tables 3 and 4: the D-GAN achieves state of the art prediction performances on CIFAR-10 with the WGAN-GP variant.
> > >
> > >
> > > “GenPU has a critical issue of mode collapse, and this is why GenPU reports 1-vs-1 rather than 5-vs-5 on MNIST.”
> > >
> > > We agree with you. Fortunately, the GAN mode collapse issue has been drastically reduced with recent GAN variants. It turns out that the D-GAN framework maintains the conventional GAN architectures (one discriminator and one generator) like DCGAN, WGAN-GP and LSGAN-GP. This enables to adapt easily the proposed approach to such conventional variants. In other words, the D-GAN architecture is more practical to follow the current GAN state of the art evolution. Furthermore, fine-tuning of GANs variants hyper-parameters is not needed when these variants are used by the proposed D-GAN framework, as presented in our article.
> > >
> > >
> > > *****
> > >
> > >
> > > These answers clarify the enumerated issues.
> > >
> > >
> > >
> > > To conclude, we thank you for highlighting these interesting points of the presented article, and for taking the time to develop them, especially concerning the PU state of the art.
> > >
> > > We sincerely thank you for your instructive review.

---

> > > > ### Comment · AnonReviewer2 · 2018-11-30
> > > > **Generality of D-GAN and GenPU**
> > > >
> > > > I am not an expert on GANs, so I still cannot follow why D-GAN is easier to benefit from recent advances whereas GenPU is harder. Is it something like spectral normalization can be used in D-GAN but cannot be used in GenPU? If so, why?

---

> > > > > ### Author Response · Authors · 2018-12-05
> > > > > **Answers to your question**
> > > > >
> > > > >
> > > > > The presented section “Without discriminator batch normalization” discussing normalization techniques is also relevant for the GenPU method.
> > > > >
> > > > > Since very recently, the spectral normalization (SN) is a GAN state of the art normalization technique. For example, recent interesting GAN models like the SAGAN (“Self-Attention Generative Adversarial Network”, 2018) propose to use it.
> > > > >
> > > > > SN is applied on the network weights. So several distribution of minibatches can be manipulated by D during the training with SN. Thus both D-GAN and GenPU can benefit to this normalization technique to accelerate and stabilize their respective adversarial trainings. We are currently testing SN on the D-GAN discriminator weights.
> > > > > The discussion part will consequently also discuss in the last version the SN technique.
> > > > >
> > > > > However, we claim that the D-GAN is relatively simpler to adapt to the new standard (One D and one G) GAN variants than GenPU because in practice, the D-GAN framework simply consists in adding a loss term in the GAN discriminator loss function, without adding any additional hyper-parameters. A GAN working without batch normalization can be adapted in this way to the proposed D-GAN framework without hyper-parameters tuning. When several changes appear in the recent GAN variants, it is more practical, from an implementation point of view, to adapt a GAN variant to the D-GAN framework than doing the contrary. This is harder to do with the GenPU algorithm implementation. For example, the SAGAN adds self-attention layers operations. The progrGAN (“Progressive growing of GANs for improved Quality, Stability, and Variation”, ICLR 2018) changes drastically the GAN training implementation.
> > > > >
> > > > > Concerning the computational cost. If we suppose that both GenPU and D-GAN need the same number of training epoch iterations to converge, then training five learning models instead of two is 2.5 times more computational demanding. This becomes a problem on high dimensional data. For example, the progrGAN (one D and one G) training time for CelebA-HQ is two weeks with one Tesla V100 GPU. If a user intends to do some GAN-based PU prototyping tests, then using two learning models instead of five is very interesting.
> > > > >
> > > > >
> > > > > We sincerely hope these answers clarify some motivations of the proposed approach. Do not hesitate to highlight what remains unclear. This helps us a lot to better communicate the key ideas of the proposed approach.
> > > > >
> > > > > We thank you for your constructive comments.

---

> > ### Comment · AnonReviewer2 · 2018-11-30
> > **Seems the authors misunderstood my point in the review**
> >
> > I know the goal of two stage PU---it is anyway similar to GenPU. I just cannot comprehensively follow the logic of your use of GANs, because Figure 1 is unclear to me. In PU learning, you cannot apply GANs without any modification, otherwise you can only generate P or U data that is not your goal.
> >
> > In GenPU, there are 2 Gs and 3 Ds, and experimentally this is the minimum number if we want to also generate P data; it can be reduced to 1G and 2Ds where 1D ensures that realP + genN approximates realU and 1D ensures that genN doesn't approximate realP (I am not an author of GenPU). In both ways, GenPU introduced the cluster assumption that is not assumed in other GANs (see Figure 2 of GenPU where p_{gn}(x) consistently has smaller supports than p_n(x), and the assumption "p_{gn}(x) almost never overlaps with p_p(x)" in their theoretical analysis). I personally think this idea of using GANs is intuitively for PU learning.
> >
> > On the other hand, D-GAN has 1G and 1D and I would like the authors to explain in the introduction why 1D is enough for identifying p_n(x) from p(x) given p_p(x) where p(x) and p_p(x) are only approximately known via data. The authors only given some equations without intuition/motivation and they are still unclear to me. Note that in PU learning, there is no PU *risk* and the expected risk is shared by PN learning and PU learning; instead, PN and PU learning have their own *risk estimators* that can estimate the unique expected risk from PN data and PU data, respectively. So what do you mean by defining a new expected risk as a performance measure to be minimized?

---

> > > ### Author Response · Authors · 2018-12-05
> > > **Answers to your questions**
> > >
> > >
> > > ***
> > > 1st paragraph answer:
> > >
> > >
> > > The original GAN (Goodfellow et al., 2014) discriminator gets in input minibatches of unlabeled examples and generated examples.
> > > Concerning the proposed approach, as illustrated in the figure 1 (updated version), the D-GAN discriminator gets in input minibatches of unlabeled examples, generated examples, and positive labeled examples.
> > >
> > > The discriminator D and the generator G of the original GAN (Goodfellow et al., 2014) are alternately trained as follow:
> > >       -	D is trained to predict in output the value “1” for unlabeled examples, and the value “0” for the generated examples.
> > >       -	G is trained to generate examples which are considered as real by D. So the generator is trained to generate examples for which D predicts the output value “1”.
> > >     => Consequently, G learns to reproduce the unlabeled examples distribution p_U.
> > >
> > > Concerning the proposed D-GAN approach, the discriminator D and the generator G are alternately trained as follow:
> > >       -	D is trained to predict in output the value “1” for unlabeled examples, and the value “0” for both generated examples and positive labeled examples.
> > >       -	G is still trained to generate examples for which the discriminator predicts the output value “1”.
> > > As highlighted in the paragraph just below the equation 1 in the updated version, we recall that if D is trained to predict in output the value “1” for unlabeled examples, and the value “0” for positive labeled examples, then D is in fact trained to exclusively predict the output value “1” for the counter-examples (included in the unlabeled dataset).
> > >     => Consequently, G exclusively learns to reproduce the counter-examples distribution p_N included in the unlabeled distribution p_U.
> > >
> > > In order to produce this new behavior in practice, we propose to add the loss function term “Ep[log(1-D(Xp))]” in the original GAN discriminator loss function (see equation 2). This is justified intuitively in the updated version of the article just below the equation 2.
> > >
> > >
> > >
> > > ***
> > > 2nd paragraph answer:
> > >
> > >
> > > The proposed approach does not follow the same intuition as the GenPU method.
> > > The proposed D-GAN approach is based on the original GAN implementation introduced by (Goodfellow et al., 2014). The original GAN implementation consists in training consecutively alternately D and G. So, we propose in our article to decompose the adversarial training such that we firstly analyze the discriminator behavior independently to the generator behavior. Then, we deduce the generator behavior which is naturally influenced by the discriminator behavior.
> > >
> > >
> > >
> > > ***
> > > 3rd paragraph answer:
> > >
> > >
> > > In PN classification, we train a learning model by optimizing a risk taking into consideration positive and negative examples.
> > > In PU classification, we train a learning model by optimizing a risk taking into consideration positive and unlabeled examples. This is simply the reason why we used the expression “PU risk” to introduce the risk presented in the equation 1.
> > > The presented risk (equation 1) is simple and does not represent by itself our contribution. However, we do a brief analysis to deduce what is the behavior of a learning model trained with this risk. Then our contribution is to propose to introduce this behavior in the original GAN discriminator loss function such that it enables G to learn the counter-examples distribution.

---

### Official Review · AnonReviewer1 · 2018-11-02
**Problem and framework not well explained**

**Rating:** 5
**Confidence:** 1

**Review:**

The motivation of the work is not clear but the novelty seems to be present.

The paper is very hard to follow as the problem description and intuition of the D-GAN is not clearly written.

Based on the experiments, the proposed method achieves marginal improvement in terms of F1 score but sometimes also slightly lower performance than other GAN based such as PGAN, so the impact of this work to solve positive unlabelled data problem is not evident.

I am personally not as familiar with the PU problem and existing frameworks so my confidence in the assessment is low; my main experience is in the computer vision for autonomous driving and sparse coding.

But my feeling is this paper is marginally below the threshold of acceptance.

---

> ### Author Response · Authors · 2018-11-25
> **Answers (Part 1)**
>
> Thanks for your review,
>
> Firstly we apologize for not making the text clear enough.
> We hope the following answers to your respective points will clarify the proposed contributions.
>
>
> ***
>
> “The motivation of the work is not clear”
>
> Motivations can be expressed as follow.
> 1: Overcome the previous state of the art approaches disadvantages.
>       -	GenPU architecture is more computational demanding (three discriminators and two generators) than standard GAN architectures (one discriminator and one generator). Furthermore, GenPU requires prior knowledge and additional loss function hyper-parameters.
>       -	The PGAN method has overfitting issues on simple datasets (see figure 6. (b)) because its approach is based on GANs imperfections.
>
> 2: A framework easily adaptable to GANs variants.
>       -	A GAN PU framework similar to standard GAN could enable a better adaptability to last and potentially future GANs variants. It is an important point because the state of the art is updated continuously but the architectures remain similar (one generator and one discriminator).
>
> 3: Adversarial training of GAN-based approaches enables to learn automatically relevant high level feature metrics.
>       -	GANs generate semantically realistic images. The most interesting aspect is probably that the error computed to evaluate the generated images quality is estimated from a high level feature point of view: the discriminator output. In this way, GANs enable relevant data augmentation.
>
>
> ***
>
> “the novelty seems to be present”
>
> The two article contributions can be highlighted as follow.
>
>
> Contribution 1: We propose to incorporate a PU risk inside the discriminator loss function.
>
> We show that a GAN can solve by itself a Positive Unlabeled learning task if the problem is well formulated: We combine a PU risk with the GAN discriminator loss function. That enables the G convergence to the distribution of counter-examples included in the unlabeled dataset.
>
> Previous GAN-based PU approaches do not include the PU risk in the discriminator cost function. GenPU and PGAN logics are as follow:
>       -	GenPU convergence is inspired by the original GAN convergence exposed by GoodFellow in 2014. The main idea is in this sentence: “Du is aimed at separating the unlabelled training samples from the fake samples of both Gp and Gn”. Thus the global system GenPU enables the convergence “pi Pgp + (1-pi) Pgn -> Pu”, with Pgp the distribution of positive samples generated by Gp, Pgn the distribution of the negative samples generated by Gn, and Pu the distribution of unlabeled samples. However, the same reasoning can be expressed using one single generator Gn if we replace the generated positive samples by the positive labeled samples that we have in a PU dataset. Thus training five different models is not necessary to address standard PU learning challenge where we have enough positive samples. This reasoning is different to the propose one.
>       -	PGAN is trained to converge towards the unlabeled dataset distribution during the first step. The PGAN exploits GANs imperfections such that the generated distribution at the adversarial equilibrium is still separable from the unlabeled samples distribution by a classifier. The PGAN does not use a PU learning risk to train its GAN part.
>
>
> Contribution 2: Highlight of a critical normalization issue discussed in the context of the proposed framework
>
> Batch-normalization (BN) technique cannot be used when several minibatches distributions (unlabeled, positive, and generated ones) are used to train a learning model.
>
> With BN, a classifier prediction for a given sample is critically influenced by the other samples of the same minibatch. As presented in the article, the consequence with a PU learning risk is that BN does not allow a classifier to distinguish positive from negative samples (see figure 5(a) and subsections 2.3 and 3.1). These sections include the analysis of this BN effect and alternative normalization solutions, such that this effect disappears.
>
> In practice, a D-GAN using BN converges towards the unlabeled samples distribution. Without, it converges exclusively towards the negative samples distribution. The normalization training impact is clearly highlighted in the figure 5.
> To the best of our knowledge, we are the first to highlight this critical phenomenon for the PU learning task.
>
>
> ***
>
> “ intuition of the D-GAN is not clearly written.”
>
> The D-GAN intuition can be expressed as follow. The discriminator D addresses to the generator G the riddle:
> “Show me what IS unlabeled AND NOT positive.”
> It turns out that negative samples included in the unlabeled dataset are both unlabeled and not positive. Consequently G addresses this riddle by learning to show the negative samples distribution to D.

---

> > ### Author Response · Authors · 2018-11-25
> > **Answers (Part 2)**
> >
> >
> > ***
> >
> > “Based on the experiments, the proposed method achieves marginal improvement in terms of F1 score but sometimes also slightly lower performance than other GAN based such as PGAN, so the impact of this work to solve positive unlabelled data problem is not evident.“
> >
> > The proposed approach overcomes previous GAN-based PU methods issues:
> >      -	GenPU issues: GenPU is not easily adaptable to the current GAN state of the art evolutions because of its untraditional adversarial framework. Moreover, GenPU uses prior knowledge. This is unpractical for example on some real incremental application datasets in which the fraction pi value can change continuously at each new training minibatch.
> >      -	PGAN issue: It has a first stage overfitting problem when it is applied on relatively simple datasets as MNIST. In fact, it is mentioned in their article: “It is also known that a GAN is not perfect in its operation when it is applied to high dimensional data, … Thus it is possible to estimate the non-zero distance d computed into the cost function of Db”. In other words, the PGAN exploits the GANs convergence defaults to address the PU learning problem.
> >
> > Globally this improvement is present, such that:
> >     -	D-GAN outperforms 75% of the time PGAN and RP methods on table 3.
> >
> > Numerically the gap is not very important, but the experimental results are consistent with the expected behavior mentioned in the method section such that the D-GAN:
> >      -	generates relevant counter-examples while preserving the standard GAN architecture;
> >      -	achieves good predictions without using prior knowledge;
> >      -	converges towards the PGAN performance for complex tasks (One vs. Rest mode on CIFAR-10): 75% of the time, the D-GAN gets prediction performances slightly above the PGAN method on CIFAR-10;
> >      -	easily outperforms in practice PGAN on simple datasets (MNIST). The overfitting issue of the PGAN for simple tasks is reduced, as illustrated on the figure 6 (b).
> >
> > Moreover, we will indicate that we did not need to fine-tune GAN variants hyper-parameters to get these results.
> >
> >
> > ***
> >
> > “my main experience is in the computer vision for autonomous driving”
> >
> > Another motivation of the proposed approach can be linked to your area.
> >
> >
> > Motivation 4: Easier incremental learning applications
> > For real applications like autonomous driving, recording unlabeled data is much more accessible than getting ground truth labeled data. Under the assumption that unlabeled data contain relevant counter-examples, using a PU learning method enables to focus the training dataset labeling effort exclusively on the samples of our class of interest.
> >
> > If the unlabeled training dataset recorded is exploited incrementally, then the fraction of unlabeled positive samples can change at each new unlabeled minibatch. For example, if the positive class is “pedestrian”, then their proportion can drastically vary from a street to another one. In this context, PU learning methods using the unlabeled dataset prior knowledge are not suitable.
> >
> > PU methods not using prior knowledge like the proposed approach can solve such problems.
> >
> >
> > ***
> >
> >
> > To sum up:
> >      -	four different motivations where presented to justify this work;
> >      -	the intuition behind the D-GAN method has been clarified;
> >      -	This article presents both scientific contributions;
> >      -	The presented results demonstrate the proposed approach ability to overcome previous GAN-based PU issues.
> >
> >
> >
> > We hope the mentioned problems have been clarified in these answers. Moreover, this helped us to improve the article understanding.
> >
> > We sincerely thank you for your constructive review.

---

### Official Review · AnonReviewer3 · 2018-11-02
**Clear Rejection**

**Rating:** 3
**Confidence:** 4

**Review:**

[Summary]
PU learning is the problem of learning a binary classifier given labelled data from the positive class and unlabelled data from both the classes. The authors propose a new  GAN architecture in this paper called the Divergent Gan (DGAN) which they claim has the benefits of two previous GAN architectures proposed for PU learning: The GenPU method and the Positive-Gan architecture. The key-equation of the paper is (5) which essentially adds an additional loss term to the GAN objective to encourage the generator to generate samples from the negative class and not from the positive class. The proposed method is validated through experiments on CIFAR and MNIST.

[Pros]
1. The problem of PU learning is interesting.
2. The experimental results on CIFAR/MNIST suggest that some method that the authors coded worked at par with existing methods.

[Cons]
1. The quality of the writeup is quite bad and a large number of critical sentences are unclear. E.g.
a. [From Abstract] It keeps the light adversarial architecture of the PGAN method, with **a better robustness counter the varying images complexity**, while simultaneously allowing the same functionalities as the GenPU method, like the generation of relevant counter-examples.
b. Equation (3) and (4) which are unclear in defining R_{PN}(D, δ)
c. Equation (6) which says log[1 - D(Xp)] = Yp log[D(Xp)] + (1-Yp) log[1-D(Xp)] which does not make any sense.
d. The distinction between the true data distribution and the distribution hallucinated by the the generator is not maintained in the paper. In key places the authors mix one with the other such as the statement that supp(Pp (Xp )) ∩ supp(Pn (Xn )) → ∅
In short even after a careful reading it is not clear exactly what is the method that the authors are proposing.

2. Section 2.2 on noisy-label learning is only tangentially related to the paper and seems more like  a space filler.

3. The experimental results in Table 4 and Table 3 do not compare to GenPU. Although the authors claim several times that the GenPU method is *onerous*, it is not clear why GenPU is so much more onerous in comparison to other GAN based methods which all require careful hyper-parameter tuning and expensive training. Furthermore the reference PN method performs worse than other PU learning methods which does not make sense. Because of this I am not quite convinced by the experiments.

---

> ### Author Response · Authors · 2018-11-19
> **Answers**
>
> Thanks for your constructive review,
>
>
> 1.
> a. “a better robustness counter the varying images complexity” will be replaced by “a better adaptability to the images complexity”
> According to the PGAN article, PGAN should not be used for simple tasks. D-GAN works on both simple and complex tasks: D-GAN is more adaptable to the images complexity.
>
> b. l(D(X), δ)  will be replaced by l(D(X), y=δ) and “⇔” by “=”.
> δ is the label of positive samples: δ substitutes both contradictory labels “0” and “1” associated to Pp in the risk Rpu (equation 1). Unlabeled positive samples are separated from unlabeled negative ones with D trained with the risk Rpu.
>
> c. Equation (6) will be replaced as follow.
> The loss function Ld of D is defined as:
> Ld = Rpu + Eg [ l(D(Xg),Yg=0) ],
> with Rpu the PU risk (equation(1)), Yg=0 the label associated to the samples Xg generated by G; Xg = G(z), and Eg the expectation for samples Xg.
> We recall Rpu = Eu [ l(D(Xu),Yu=1) ] + Ep [ l(D(Xp),Yp=0) ], with Yu=1 the label of unlabeled samples Xu with the expectation Eu, and Yp=0 the label of labeled positive samples Xp with the expectation Ep.
> The loss function “l” used in the proposed D-GAN framework can be the binary cross-entropy “H” such that “l=-H”. So:
> Ld = Eu [ -H(D(Xu),Yu=1) ] + Ep [ -H(D(Xp),Yp=0) ] + Eg [ -H(D(Xg),Yg=0) ].
> The binary cross-entropy H is defined as below:
> H(D(X),Y) = - Y log(D(X)) – (1-Y) log(1-D(X)), with Y represents the label associated to the D input samples X. Thus H(D(X),Y=1) = - log(D(X)) and H(D(X),Y=0) = - log(1-D(X)).
> Finally, Ld can be developed as follow:
> Ld = Eu [ log[D(Xu)] ] + Ep [ log[1-D(Xp)] ] + Eg [ log[1-D(Xg)] ].
> This shows the incorporation of Rpu (equation(1)) inside the D-GAN loss function (equation(5)).
> The role of G during the adversarial training is to generate samples considered by D as “1”. Only negative samples are considered as “1” by D thanks to the Rpu risk. This justifies intuitively the G convergence towards the negative samples distribution.
>
> d. The sentence part "such that we have supp(Pp (Xp )) ∩ supp(Pn (Xn )) → ∅, with supp the support function of probability distributions." will be removed.
> We talked about D ability to distinguish positive samples distribution Pp from negative one Pn. If D does this distinction, then G converges towards Pn. If D fails to do this task, then G converges to the unlabeled samples distribution Pu as the PGAN.
>
> 2. We take into consideration your comment. This is not the main message of the article. Section 2.2 will be removed from the method part.
>
> 3.
> “results in Table 4 and Table 3 do not compare to GenPU.”
> We do not compare our results to GenPU method for the challenging One vs. Rest task because:
> a. GenPU method is not reproducible.
>     - Code not provided.
>     - Implementation details are missing in their article: Three hyper-parameters (lambda_P, lambda_N and lambda_U) are introduced in the GenPU article, but the values are not specified. They are important for the GenPU training with respect to their role inside the GenPU cost function (GenPU equation (3)): Instructions 8, 9 and 10 of the GenPU pseudo-code (GenPU Algorithm 1) apply them directly to the prior knowledge parameters πp and πn (=1-πp). That makes impossible the GenPU reproducibility.
>
> b. GenPU mode collapse issue does not enable to perform complex tasks as the One vs. Rest challenge.
>
> c. GenPU is not valorized in their article as an interesting alternative for the standard PU context where we own relatively enough positive labeled samples: GenPU article does not present results with more than 100 positive labeled samples.
>
> d. The goal of tables 3 and 4 is to compare methods which do not need prior knowledge.
>
>
> “the authors claim several times that the GenPU method is *onerous*”
> GenPU training computational cost cannot be quantified: GenPU is not reproducible and training epoch iterations needed to converge are not specified. If we consider that both standard GAN and GenPU architectures need the same number of training epochs to converge to the expected distribution, then training five models (GenPU) instead of two (D-GAN) is more computational demanding.
> D-GAN does not add or modify hyper-parameters of GAN variants tested (GAN, DCGAN, WGAN-GP, LS-GAN).
>
>
> “the reference PN method performs worse than other PU learning methods which does not make sense.”
> D-GAN performs better than PN on CIFAR-10 because:
>     - It learns relevant counter-examples distribution. RP article discusses the same behavior on CIFAR-10.
>
>     - Generated images enable data augmentation. GANs latent linear interpolations result in semantic images interpolations outputs. Thus GANs learn generic representation.
> It is not observed on MNIST because data augmentation is difficult to produce on low-dimensional data.
> PGAN score when πp=0 (as for PN) will be added in tables 3 and 4 to highlight this effect.
>
> This phenomenon is not straightforward, but these reasons clarify it.
>
>
> We sincerely thank you for your review.

---

> > ### Author Response · Authors · 2018-11-19
> > **Furthermore,**
> >
> >
> >
> > ***
> > PGAN score when πp=0 (annoted as PNGAN data augmentation reference) is now indicated in tables 3 and 4 to highlight the GAN effect on CIFAR-10.
> > => WGAN "data augmentation" increases the reference PN average F1-Score on CIFAR-10 from 0.68 to 0.812.
> > ***
> >
> > - Another novelty of the presented article is to highlight a critical Batch-Normalization effect on the discriminator (sections 2.3 and 3.1).
> >
> >
> > - D-GAN intuition can be expressed as follow:
> >         “- Show me what IS unlabeled AND NOT positive.”
> > This is the task asked by D to G. Negative samples are both unlabeled and not positive. Consequently G learns to show the negative samples distribution to D.
> >
> > This article presents an interesting contribution by merging GANs and PU learning areas in this way.
> >
> > ***
> >
> > Your review helped us to clarify some formulations of the proposed method. You also highlighted that the article omitted some justifications concerning the experimental results. We apologize for not making the text clear enough. We will use shorter and concise sentences in the article.
> >
> > These previous answers to your respective points will contribute to improving the presentation clarity and strengthening the experiments.
> >
> > We sincerely thank you for your review.

---

### Comment · AnonReviewer3 · 2018-10-25
**Unclear definitions**

This paper is studying the problem of PU learning which is an important and interesting problem, however I am having difficulty in reading the paper key definitions and equations are badly written. Please clarify the following:

a) Equation (6) says log[1 - D(Xp)] = Yp log[D(Xp)] + (1-Yp) log[1-D(Xp)] which does not make any sense. What are the authors saying here?

b) Equation (3) defines R_{PN}(D, δ) in terms of l(D(X), δ) but l(D(X), δ) is not defined properly in equation (4). The left hand side of (4) has l(D(X), δ)  but δ vanishes on the right hand side of that equation. I have no idea what is going on here.

c) The authors frequently confuse the true data distribution and the distribution hallucinated by the the generator. For example consider the expressions that "supp(Pp (Xp )) ∩ supp(Pn (Xn )) → ∅ " Which distribution are the authors talking about? Is it an assumption on the true data distribution required for learning ? or this is a property of the generator's distribution.

D) The experimental results in Table 4 and Table 3 do not compare to GenPU. Although the authors claim several times that the GenPU method is *onerous*, it is not clear why GenPU is so much more onerous in comparison to other GAN based methods which all require careful hyper-parameter tuning and expensive training. Furthermore the reference PN method performs significantly worse than other PU learning methods which does not make sense. The PN method should be much better or comparable to the performance of any PU method. Please clarify.

---

### Meta-Review · Area_Chair1 · 2018-12-13
**Proposed model targets an interesting problem but paper could need a bit more work**

**Confidence:** 4
**Recommendation:** Reject

**Metareview:**

With positive unlabeled learning the paper targets an interesting problem and proposes a new GAN based method to tackle it. All reviewers however agree that the write-up and the motivation behind the method could be made more clear and that novelty compared to other GAN based methods is limited. Also the experimental analysis does not show a strong clear performance advantage over existing models.